# Phase Behavior and DFT Calculations of Laterally Methyl Supramolecular Hydrogen-Bonding Complexes

**Hoda A. Ahmed [1,2,*], Mohamed Hagar [1,3,*]**  **and Omaima A. Alhaddad [4]**

1   Chemistry Department, College of Sciences, Yanbu, Taibah University, Yanbu 30799 Saudi Arabia
2   Faculty of Science, Department of Chemistry, Cairo University, Cairo 12613, Egypt
3   Faculty of Science, Chemistry Department, Alexandria University, Alexandria 21321, Egypt
4   hemistry Department, College of Sciences, Al-Madina Al-Munawarah, Taibah University, Al-Madina 30002, Saudi Arabia; OHADDAD@taibahu.edu.sa
*   Correspondence: ahoda@sci.cu.edu.eg (H.A.A.); mohamedhaggar@gmail.com (M.H.); Tel.: +966542015471 (H.A.A.); +966545527958 (M.H.)

**Abstract:** Four new series of laterally methyl-substituted hydrogen-bonded supramolecular complexes were prepared. The prepared complexes were thermally investigated by differential scanning calorimetry (DSC) and phases identified by polarized light microscopy (PLM). Supramolecular hydrogen-bonded complexes formed from a 1:1 mixture of any two derivatives, bearing different alkoxy chains, of 4-alkoxyphenylazobenzoic acid and 4-(2-(pyridin-4-yl)diazenyl-(2-(or 3-)methylphenyl) 4-alkoxybenzoate. The investigated 1:1 mixture made by introducing a lateral methyl group by different spatial orientation angles into pyridine-based components. All new complexes were confirmed by Fourier-transform infrared spectroscopy (FTIR) and computational calculations used to study their stabilities. It is found that the prepared complexes are dimorphic, exhibiting smectic C and enhanced nematic phases. A comparison was made between the new series and previously investigated simpler complexes, revealed that the incorporation of the phenylazo group elongate the mesogenic part and hence broad nematic phases were obtained with high stability.

**Keywords:** supramolecular-hydrogen bonding; lateral methyl; 4-alkoxyphenylazobenzoic acid; nematic stability; computational calculations; DFT

## 1. Introduction

Rod-like liquid crystalline compounds have been intensively studied for technological and scientific important applications such as sensing, optoelectronic display and biologically active materials [1–5]. Mixtures and especially eutectic composition of liquid crystals are very suitable for device applications. It has been reported that substitution in the lateral position decreases the thermal stability of both solid states as well as the mesophase due to the steric effects and the intramolecular hindering of the lateral group [6–13]. Moreover, the computational methods [14–24] have been used as an excellent tool for designing a molecule with projected properties. Many research articles have reported different synthetic methodologies for the preparation of new liquid crystalline materials. Enhanced performance of liquid crystals resulted in the formation of hydrogen bonding (HB) in standard ferroelectric materials [25]. The induction of supramolecular [26–32] design into liquid crystals (LCs) by the involvement of covalent interactions between different chemical moieties opened a new field of research. Many binary hydrogen-bonded systems, based on derivatives of carboxylic acids as the hydrogen donors and pyridine as an acceptor, have been extensively studied and reviewed [33–36] Recently, nematic, smectic and columnar mesophases have been induced by hydrogen-bonding between pyridines and benzoic acids [37–40]. Furthermore, the ability for

*cis-trans*-isomerisation of the azobenzene unite by its irradiation with Ultra violet (UV) light makes it of special interest in many applications. The lengthening of the rigid-rod core of azo-based liquid crystals by the formation of a hydrogen bonding supramolecular complex that may not occur in the individual components highly modifies their properties [41,42]. Moreover, changing the polarity and/or polarizability of both hydrogen bonded components affects the strength of formed hydrogen bond and consequently induces the liquid crystalline character of the interacting components.

Continuing our interest, herein, we have investigated the change of the thermal behavior due to intermolecular hydrogen-bond formation between the 4-alkoxyphenylazo-benzoic acids [28] **I***n*, and 4-(2-(pyridin-4-yl)diazenyl-(2-(or 3-)methylphenyl) 4-alkoxybenzoate, **II***m* and **III***m* [33,34] respectively. The second aim is to illustrate the decrease in the melting temperature of the prepared supramolecular complexes by introducing a methyl group into the lateral position (2 or 3) of the central ring of the pyridine-based part and using the Density functional theory (DFT) computational calculations to study their stability in different spatial orientation of the methyl group, which is oriented at different angles (Schemes 1 and 2). A third aim is to study the effect of introducing of phenyl azo unit to the 1:1 supramolecular complex previously reported [33,34], in comparison with those of hydrogen-bond formation between 4-alkoxybenzoic acids (**IV***n*) as proton donors and 4-(2-(pyridin-4-yl)diazenyl-(2-(or 3-)methylphenyl) 4-alkoxybenzoate (**II***m* and **III***m*) as acceptors (**IV***n*/**II***m* and **IV***n*/**III***m*).

**Scheme 1.** 1:1 supramolecular hydrogen-bonded complexes, **I***n*/**II***m*.

**Scheme 2.** 1:1 supramolecular hydrogen-bonded complexes, **I***n*/**III***m*.

## 2. Experimental

DCC (N,N′-dicyclohexylcarbodiimide) and DMAP (4-dimethylaminopyridine) and all solvents were obtained from Aldrich (Wisconsin, WI, USA).

Calorimetric measurements were recorded on TA Instruments Co. Q20 differential scanning calorimeter (DSC) (New Castle, DE, USA). Small samples (2–3 mg) placed in aluminum pans under a nitrogen gas atmosphere at a heating rate of 10 °C/min. The DSC was calibrated using indium and lead according to their melting temperature and enthalpy.

DSC was used in the determination of the transition temperatures for all components and their H-bonded complexes (**I***n*/**II***m* and **I***n*/**III***m*). However, polarized light microscopy (PLM) (Olympus, Wild, Germany) equipped with Mettler FP82HT hot stage was used in the identification of the mesophase type.

### 2.1. Supramolecular Complexes Preparation

Supramolecular complexes (**I**n**/II**m and **I**n**/III**m), in 1:1 molar ratios were prepared by melting the appropriate amounts of each component, stirring to give an intimate blend and then cooling with stirring to room temperature (Scheme 3).

**I**n, *n* = 6, 8, 10 ,12

+

**II**m (2-CH$_3$),**III**m (3-CH$_3$) *m* = 8 and 16

**I**n/ **II**m *or* **I**n/ **III**m

**Scheme 3.** Preparation of 1:1 supramolecular hydrogen-bonded complexes (**I**n**/II**m and **I**n**/III**m).

### 2.2. Characterizations

The purity of the individual compounds was checked with silica gel thin-layer chromatography TLC (E. Merck. Their molecular formulae were recognized by the elemental analyses (Thermo Scientific Flash 2000 CHS/O Elemental Analyzer, Milan, Italy), infrared (Perkin-Elmer, Inc., Shelton, CT, USA), and [1]H-NMR spectroscopy (Varian EM350L 300 MHz spectrometer, Oxford, UK). The results agreed with those reported in the literature [28,33,34].

DSC and Fourier transform infrared spectroscopy (FTIR, Nicolet iS 10 Thermo Scientific, Waltham, MA, USA) were used for confirmation of the formation of the supramolecular complexes (**I**n**/II**m and **I**n**/III**m).

### 2.3. Computational Methods and Calculations

Gaussian 09 software (2009, Gaussian. Inc.: Wallingford, CT, USA) [43] using the DFT/B3LYP method [24,44] using 6-31G (d,p) basis set was used for the theoretical calculations at 298 K for all compounds under investigation. Gauss View [45] was used for the structures drawing. Similarly, the frequencies calculated were carried out at the same level as the method. The frequency calculations showed that all structures were stationary points in the geometry optimization method with none imaginary frequencies.

## 3. Results and Discussion

### 3.1. Phase Behavior of 1:1 Molar Supramolecular Complexes

The mesophase behavior of the present 1:1 supramolecular complex **I***n*/**II***m* and **I***n*/**III***m* was investigated by DSC and PLM. Textures observations by PLM were verified by the DSC measurements and mesophase types were identified for all prepared supramolecular complexes.

It should be noted that the phase behavior of the prepared 4-alkoxyphenylazobenzoic acids **I***n* exhibit a wide range of stability in the smectic C phase (SmC) and the nematic phase (N) with a narrow stability range, [28] while the lateral methyle azopyridines **II***m* and **III***m* exhibit very small mesophase stability in the smectic C phase [33,34]. Therefore, it was interesting to investigate the phase behavior by introducing a polar lateral group to the supramolecular complexes resulting from mixing compounds **I***n*/**II***m* and **I***n*/**III***m* to study the spatial orientation of the lateral methyl group on the mesophase behavior of the prepared complexes.

Table 1 collects the transition temperatures and their corresponding enthalpy of transitions for the 1:1 molar ratio of complexes **I***n*/**II***m* and **I***n*/**III***m*. Transition temperatures were graphically related with the alkoxy chain length (*n*) of the acid component shown in Figures 1 and 2, respectively. As can be seen from Table 1 and Figures 1 and 2, all new complexes are exhibit enantiotropic smectic C and nematic mesophases in addition to two crystalline phases (Cr1 and Cr2). In addition, the nematic stability is shows a great enhancement at *n* = 10. The SmC mesophase range is increasing with the acid chain length (*n*). Representative textures of the mesophases under a polarizing optical microscope are shown in Figure 3.

**Table 1.** Phase transition temperatures (°C), enthalpy of transitions (kJ/mol) for the supramolecular complexes **I***n*/**II***m* and **I***n*/**III***m*.

| System | $T_{Cr1\text{-}Cr2}$ | $T_{Cr2\text{-}C}$ | $\Delta H_{Cr2\text{-}C}$ | $T_{C\text{-}N}$ | $\Delta H_{C\text{-}N}$ | $T_{N\text{-}I}$ | $\Delta H_{N\text{-}I}$ |
|---|---|---|---|---|---|---|---|
| **I6/II8** | 76.3 | 96.4 | 34.5 | 126.2 | 3.1 | 166.6 | 1.2 |
| **I8/II8** | 74.8 | 96.3 | 25.8 | 137.8 | 6.5 | 177.0 | 1.7 |
| **I10/II8** | 77.3 | 96.3 | 32.7 | 137.0 | 5.6 | 184.1 | 1.9 |
| **I12/II8** | 79.1 | 95.9 | 31.9 | 139.3 | 6.9 | 170.2 | 2.8 |
| **I6/II16** | 85.0 | 97.4 | 68.6 | 128.0 | 4.6 | 177.2 | 1.6 |
| **I8/II16** | 77.1 | 94.5 | 54.5 | 138.8 | 3.9 | 179.4 | 1.1 |
| **I10/II16** | 85.7 | 99.0 | 66.4 | 137.7 | 4.1 | 184.3 | 1.4 |
| **I12/II16** | 86.3 | 104.5 | 51.9 | 140.0 | 3.7 | 170.2 | 1.2 |
| **I6/III8** | 84.5 | 94.7 | 42.6 | - | - | 171.5 | 2.1 |
| **I8/III8** | 82.8 | 93.1 | 37.4 | 135.3 | 3.1 | 177.3 | 1.2 |
| **I10/III8** | 84.6 | 95.5 | 35.4 | 136.9 | 4.6 | 184.0 | 2.6 |
| **I12/III8** | 85.4 | 94.9 | 36.5 | 139.3 | 6.6 | 170.3 | 3.2 |
| **I6/III16** | 78.0 | 95.2 | 59.3 | - | - | 175.4 | 1.8 |
| **I8/III16** | 77.1 | 95.4 | 53.7 | 139.0 | 4.7 | 176.9 | 1.3 |
| **I10/III16** | 78.1 | 95.0 | 51.7 | 137.5 | 5.2 | 184.2 | 2.6 |
| **I12/III16** | 79.2 | 98.2 | 51.3 | 139.8 | 5.3 | 170.2 | 2.8 |

Cr1-Cr2 = crystal to crystal transition; Cr2-SmC = crystal to smectic C phase; SmC-N = smectic C to Nematic transition**;** N-I = Nematic to isotropic liquid transition.

Figure 1 shows that for a given value of alkoxy chain length (*m*) of the pyridine-based complement, where the lateral $CH_3$ group is in position-2 of the central ring, the melting temperatures of the prepared mixtures are slightly affected by the length of the acid alkoxy-chains and exhibit low values. In addition, a wide range of nematic phases was exhibited for the resultant complexes **I***n*/**II***m***,** which were produced by intermolecular hydrogen bonding.

The results from Table 1 were also related to the acid alkoxy chain length (*n*) in Figure 2. This shows that**,** for a given value of *m*, where the lateral $CH_3$ group is in position-3 of the central benzene ring of the azopyridine component, low values for the melting points were observed and as usual changes

irregularly with the alkoxy-chain length of the acid. In addition, the constructed supramolecular complexes **I**n**/III**m also showed a nematic phase with relatively high range.

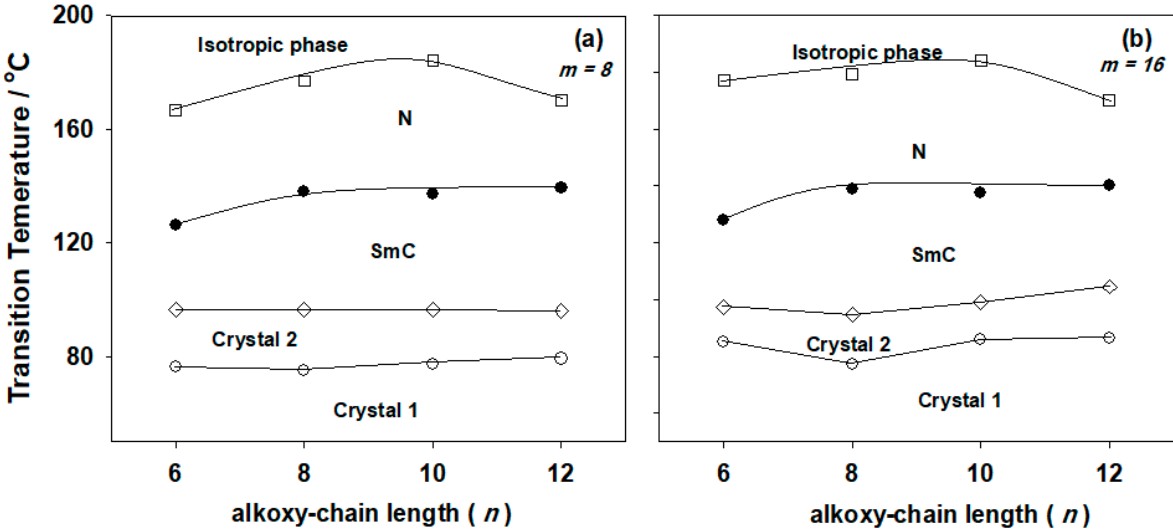

**Figure 1.** Effect of the alkoxy-chain length ($n$) of the azopyridine derivatives (**II**m) on the mesophase behavior of the 1:1 supramolecular hydrogen-bonded complexes (**a**) $m = 8$; (**b**) $m = 16$.

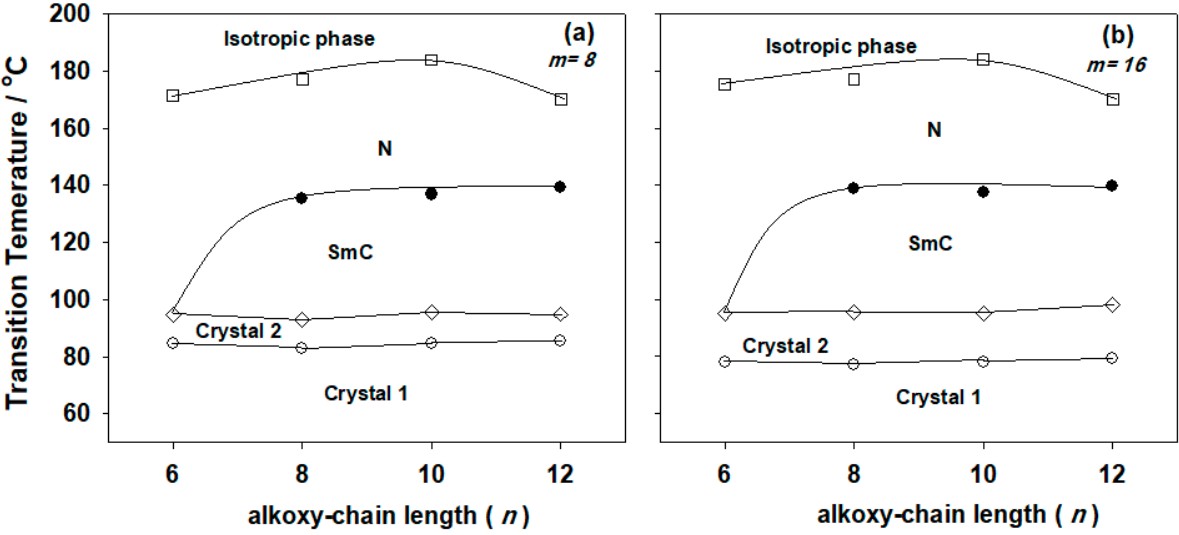

**Figure 2.** Effect of the alkoxy-chain length ($n$) of the azopyridine derivatives (**III**m) on mesophase behavior of the 1:1 supramolecular hydrogen-bonded complexes (**a**) $m = 8$; (**b**) $m = 16$.

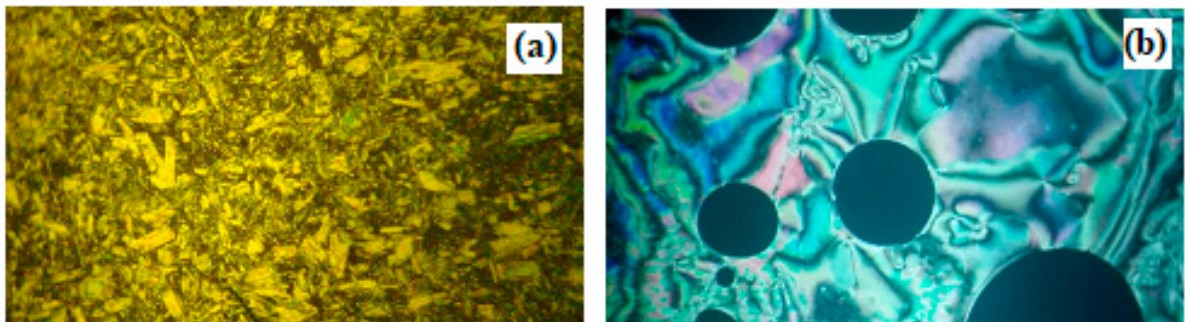

**Figure 3.** PLM textures of the supramolecular complex **I10/II8** (**a**) SmC phase at 120.0 °C; and (**b**) nematic phase at 181.0 °C.

## 3.2. Effect of Introduction Phenylazo Part to the 1:1 Supramolecular Complexes

A comparison was constructed between the mesophase stabilities ($T_C$) of 4-n-alkoxyphenylazo benzoic acids (**I***n*) supramolecular hydrogen-bonding complexes (**I***n*/**II***m*) and their corresponding 4-n-alkoxy benzoic acids (**IV***n*/**II***m*, Scheme 4), and also (**I***n*/**III***m*) and their corresponding (**IV***n*/**III***m*, Scheme 5) to study the effect of introducing an extra phenylazo moiety to molecules have 4-alkoxy benzoic acids (**IV***n*). The study related the mephase behaviour of 1:1 molar mixtures [33,34] as a function of the acid alkoxy-chain length (*n*) and is represented graphically in Figure 4. The results revealed that the elongation of the mesogenic core length by a phenylazo moiety led to an increase in the mesophase stabilities. In addition, a nematic enhancement obtained in the present new mixtures **I***n*/**II***m* and **I***n*/**III***m* almost equally upon incorporation of the phenylazo group in **IV***n*/**II***m* and **IV***n*/**III***m*, respectively.

**Scheme 4.** 1:1 supramolecular hydrogen-bonded complexes, **IV***n*/**II***m*.

**Scheme 5.** 1:1 supramolecular hydrogen-bonded complexes, **IV***n*/**III***m*.

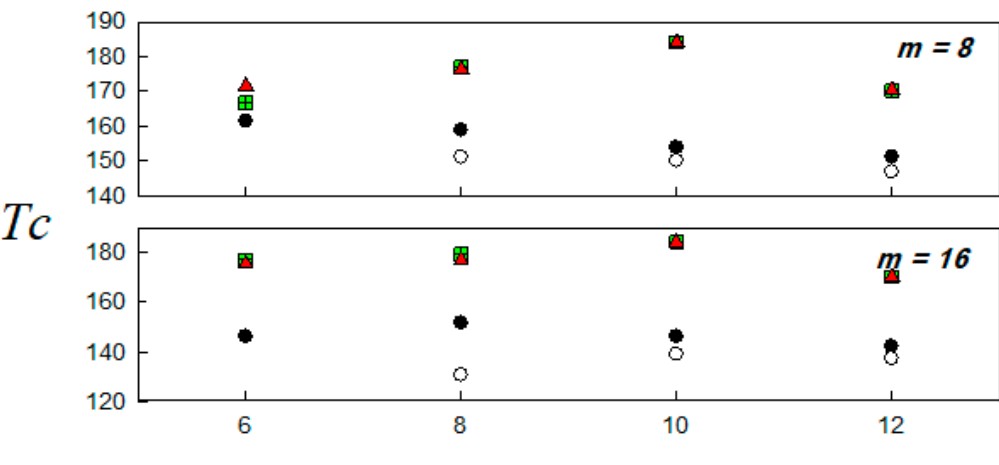

**Figure 4.** Dependence of the stability temperature (*Tc*) on the terminal alkoxy chain length (*n*) substituent of the acid moiety; I*n*/II*m* (▲); **I***n*/**III***m* (■); **IV***n*/**II***m* (●); **IV***n*/**III***m* (○).

## 3.3. DFT Calculations and Thermal Stability

The optimized molecular structure of **I12, II16, III16** and their supramolecular H-bonded complexes (**I12/II16, I12/III16**) were studied in the gas phase using DFT/B3LYP methods using a 6-31G (d,p) basis set by using GAUSSIAN 09W (Figure 5, Table S1–S5). The stability and optimized structure of the compound is confirmed by the absence of imaginary frequency for all of these compounds. As shown in the figure, the calculations predicted the non-linearity of the complexes.

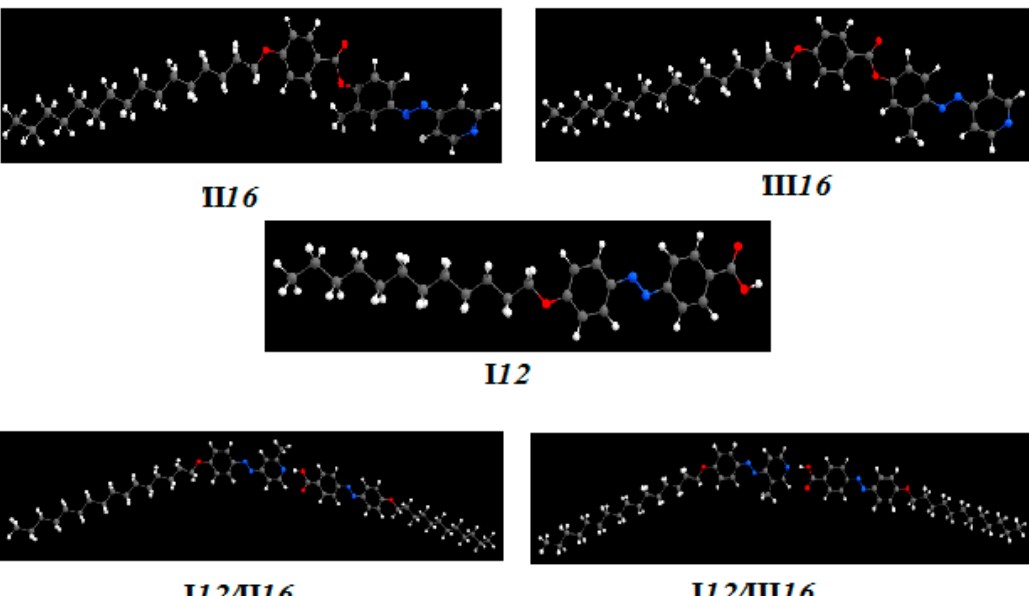

**Figure 5.** Calculated geometrical structure of **I***12*, **II***16*, **III***16*, **I***12*/**II***16* and **I***12*/**III***16*.

Table 2 illustrates the calculated total energies, dipole moments and thermodynamic functions of the studied complexes (**I***12*/**II***16*, **I***12*/**III***16*) as well as the acid (**I***12*) and the bases (**II***16*, **III***16*). DFT calculations showed the extra stability of the complex **I***12*/**II***16* with respect to **I***12*/**III***16* by 2.3657 Kcal/mol and 2.8589 Kcal/mol, enthalpy change and Gibbs free energy change, respectively. Moreover, all compounds are polar. However, the position of the methyl group affects the polarity, where the complex **I***12*/**III***16* showed more polarity than **I***12*/**II***16*, 7.7514 and 6.6169 debye, respectively.

**Table 2.** Thermal parameters (Hartree/Particle) and dipole moment (Debye).

| Parameter | | I12 | II16 | III16 | I12/II16 | I12/III16 |
|---|---|---|---|---|---|---|
| $E_{corr}$ | | 0.552074 | 0.762269 | 0.762263 | 1.220213 | 1.219892 |
| ZPVE | | −1307.701790 | −1751.112125 | −1751.111524 | −2639.339419 | −2639.343300 |
| $E_{tot}$ | | −1307.670531 | −1751.069093 | −1751.068433 | −2639.270880 | −2639.274657 |
| H | | −1307.669587 | −1751.068149 | −1751.067489 | −2639.269936 | −2639.273713 |
| G | | −1307.770916 | −1751.199293 | −1751.199207 | −2639.465599 | −2639.470155 |
| ΔH | | | | | 000000 | 0.003777 |
| ΔG | | | | | 000000 | 0.004556 |
| | X | 1.8494776 | 3.2045878 | 2.9735829 | 2.4718444 | −2.1457531 |
| Dipole | Y | −0.2220247 | −0.546798 | 0.6756452 | −0.0119251 | −0.4343364 |
| Moment | Z | 0.025892 | −0.0004635 | 0.0391378 | 0.8166558 | −0.0123549 |
| | Total | 4.7351 | 8.2629 | 7.7514 | 6.6169 | 7.7514 |

ZPVE: Sum of electronic and zero-point energies; $E_{tot}$: Sum of electronic and thermal energies; H: Sum of electronic and thermal enthalpies; G: Sum of electronic and thermal free energies.

Table 3 shows the calculated geometric bond length of the characteristic groups of **I***12*, **II***16*, **III***16*, **I***12*/**II***16* and **I***12*/**III***16*. It is worthy to note that the formation of intermolecular H- hydrogen bonding between the acid **I***12* and the bases **II***16*, **III***16* highly affects the bond length and consequently the bond strength. As shown in Table 3, the bond length of the O–H bond length (0.97588 Å) of the acid increases to be 1.04420 Å and 1.04760 Å for **I***12*/**II***16* and **I***12*/**III***16*, respectively, where the sharing of the hydrogen atom in the formation of the H-bond decreases the electron sharing with the oxygen atom to form an O–H bond, resulting in its elongation. On the other hand, the more electron donation of the carboxylic group oxygen decreases the double bond character of its C=O and consequently increases the bond length to be 1.25123 Å for **I***12*/**II***16* and 1.25104 Å for **I***12*/**III***16* instead of 1.23711 Å for the free acid. The pyridine moiety electron donation during the H-bond formation affects the C=N

and C=C of the pyridine ring in a similar manner, where the H-bond formations increase the double bond character for both (strengthened). Similarly, the azo group either for the acid moiety or the base moiety has been strengthened by decreasing the bond length.

**Table 3.** The calculated geometric bond length in Å of characteristic groups of **I12, II16, III16, I12/II16** and **I12/III16**.

| Compound | $OH_{COOH}$ | $C=O_{COOH}$ | $C-O_{COOH}$ | $C=N_{Pyr}$ | $C=C_{Pyr}$ | $N=N_{acid}$ | $N=N_{base}$ | H-bond |
|---|---|---|---|---|---|---|---|---|
| **I12** | 0.97588 | 1.23711 | 1.38408 | | | 1.28188 | | |
| **II16** | | | | 1.35599 | 1.39091 | | 1.27854 | |
| **III16** | | | | 1.35597 | 1.39088 | | 1.27988 | |
| **I12/II16** | 1.04420 | 1.25123 | 1.34683 | 1.34989 | 1.38555 | 1.28110 | 1.28236 | 1.56777 |
| **I12/III16** | 1.04760 | 1.25104 | 1.34753 | 1.35153 | 1.39348 | 1.28112 | 1.28166 | 1.57783 |

Infrared spectra for both the monomeric compounds and the supramolecular complexes have been estimated theoretically in the gas phase and experimentally in KBr at room temperature in the region of $400–4000$ $cm^{-1}$. The data are tabulated in Table 4. Experimental and DFT calculated spectral data show almost the same results and the same shift direction of peaks. However, the little difference between the experimental and the calculated results due to the single unit calculation in the gas phase without any consideration of the intermolecular interaction. As previously described, the supramolecular complex formation affects the bond strength and consequently shifting the wavelength of vibration either to blue shift or red shift. The main effect of the H-bonding formation was expected for the O–H group (from 3360 $cm^{-1}$ for **I12** to 2362 $cm^{-1}$ for **I12/II16** $cm^{-1}$ or 2304 $cm^{-1}$ for **I12/III16**), these data were consistent with the experimental results. Such decreasing in the wave number could be explained in the term of the weakness of the O–H bond by its H-bond formation. Moreover, an important evidence of intermolecular hydrogen bonding is C=O bond stretching vibration, where the lower wave number stretching vibration of C=O bond of the cyclic carboxylic acid ($\acute{v} = 1679$ $cm^{-1}$, experimentally and 1666 $cm^{-1}$, theoretically) is shifted to a higher wave number ($\acute{v} = 1682$ $cm^{-1}$, experimentally and 1668 $cm^{-1}$, theoretically) of the hydrogen-bonded dimers. Moreover, vibration peaks at 3150 $cm^{-1}$, 2545 $cm^{-1}$, 1925 $cm^{-1}$ are due to Fermi resonance of A-type, B-type, C-type, respectively. However, there is no significant peak shifting either experimentally or theoretically for $C=N_{Pyr}$, $C=C_{Pyr}$ and $N=N_{acid.}$ groups.

**Table 4.** The wave numbers theoretically (experimentally) of characteristic groups of **I12, II16, III16, I12/II16** and **I12/III16**.

| Compound | $OH_{COOH}$ | $C=O_{COOH}$ | $C-O_{COOH}$ | $C=N_{Pyr}$ | $C=C_{Pyr}$ | $N=N_{acid}$ | $N=N_{base}$ |
|---|---|---|---|---|---|---|---|
| **I12** | 3360 | 1666 (1679) | 1374 (1295) | | 1601 (1589) | 1424 (1417) | |
| **II16** | | | | 1591 (1600) | 1616 (1615) | | 1434 (1469) |
| **III16** | | | | 1590 (1595) | 1617 (1615) | | 1434 (1472) |
| **I12/II16** | 2362 (2353) | 1668 (1681) | 1319 (1298) | 1593 (1597) | 1625 | 1426 (1420) | 1428 (1469) |
| **I12/III16** | 2304 (2353) | 1672 (1681) | 1319 (1296) | 1592 (1592) | 1635 | 1426 (1420) | 1421 (1469) |

*3.4. Frontier Molecular Orbitals*

Figure 6 shows the calculated frontier molecular orbitals of **I12, II16, III16, I12/II16** and **I12/III16** at isosurface value of 0.02. Frontier molecular orbitals (FMOs) refer to the highest occupied and lowest unoccupied molecular orbital, HOMO and LUMO, respectively. The HOMO acts as an electron donor. The LUMO accepts electrons and acts as an electron acceptor. They could illustrate a sensible qualitative prediction of the ability of electron transport by the excitation of the electrons from HOMO to LUMO. The FMOs energies of the HOMO and the LUMO energies of the compounds of **I12, II16, III16, I12/II16** and **I12/III16** are given in Table 5. Based on the value of energies of the frontier molecular orbitals, various values such as chemical hardness ($\eta$) (the resistance to charge transference), and global softness (*S*) could be calculated as follow:

The hardness of the compounds:

$$\eta = (I - A)/2 \tag{1}$$

where *I* is the ionization energy, *A* is the electron affinity, $(I - A) = \Delta E(E_{LUMO} - E_{HOMO})$.

The global softness:

$$S = 1/2\eta. \tag{2}$$

The hardness measures the resistance of the compound to deform its electron cloud by small perturbation encountered during the chemical process and much less polarizable system. However, soft systems are more polarizable. As shown in Table 5, the H-bonded complexes are softer than either the acid or the bases, so they were predicted to show more polarizability than the free molecules.

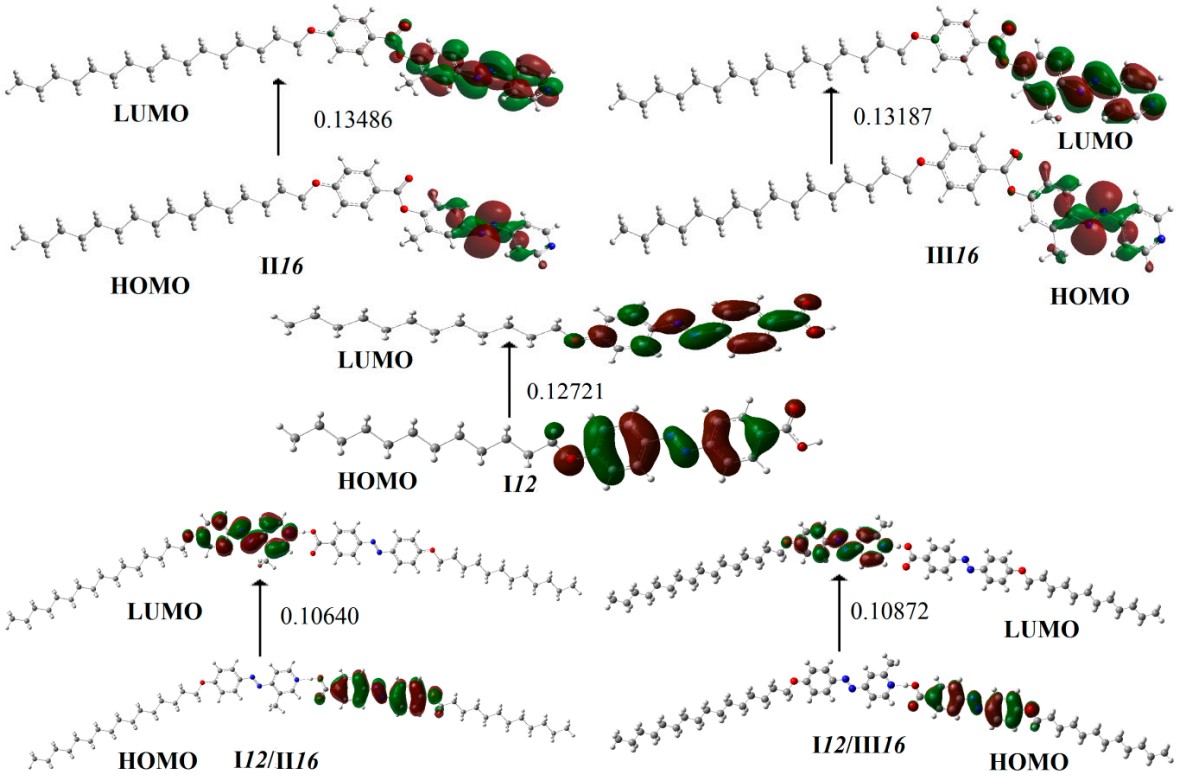

**Figure 6.** Calculated frontier molecular orbitals of **I***12*, **II***16*, **III***16*, **I***12*/**II***16* and **I***12*/**III***16*.

**Table 5.** Molecular orbital energies, hardness (*η*) and global softness (*S*) of **I***12*, **II***16*, **III***16*, **I***12*/**II***16* and **I***12*/**III***16*.

| Compound | $E_{HOMO}$ (a.u) | $E_{LUMO}$ (a.u) | $\Delta E(E_{LUMO} - E_{HOMO})$ (a.u) | $\eta = \Delta E(E_{LUMO} - E_{HOMO})/2$ | $S = 1/2\eta$ |
|---|---|---|---|---|---|
| **I***12* | −0.22959 | −0.10238 | 0.12721 | 0.063605 | 0.031803 |
| **II***16* | −0.23916 | −0.10430 | 0.13486 | 0.06743 | 0.033715 |
| **III***16* | −0.23827 | −0.10640 | 0.13187 | 0.065935 | 0.032968 |
| **I***12*/**II***16* | −0.21766 | −0.11126 | 0.10640 | 0.0532 | 0.0266 |
| **I***12*/**III***16* | −0.21814 | −0.10942 | 0.10872 | 0.05436 | 0.02718 |

### *3.5. Molecular Electrostatic Potential (MEP)*

The molecular electrostatic potential (MEP) has been calculated as shown in Figure 7. It illustrated that the blue regions show the least electron density (an electrophilic center), and the red regions indicate the highest electron density (a nucleophilic center). The carboxylic group of the azocarboxylic acid monomer is the red region with the highest electron density. On the contrary, the blue (the least electron density) regions in the H-bonded complex are mainly localized over the azopyridine monomer. This kind of charge distribution affects the dipole moment as well as molecular polarizability,

which affects many optical properties of the compounds under study [46]. Moreover, MEP is one of the best suitable tools for prediction of the presence of inter- and intramolecular interactions of the studied compounds. Moreover, we have very recently reported the relationship between the charge distribution that could be predicted by DFT calculation in term of MEP and the type of the mesophase [47–50].

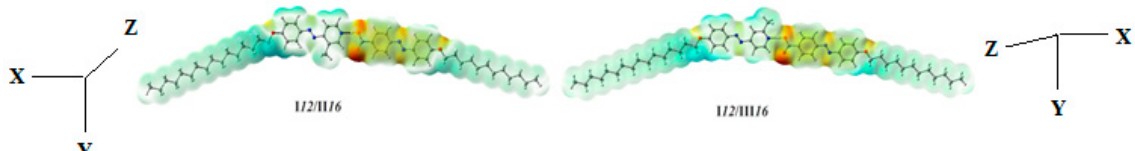

**Figure 7.** Molecular electrostatic potential (MEP) of **I12/II16** and **I12/III16.**

### 3.6. Entropy Transition Changes

The entropies of the SmC–to-N and N-to–isotropic liquid transitions were estimated for all complexes (**I**$n$/**II**$m$, **I**$n$/**III**$m$, **IV**$n$/**II**$m$ and **IV**$n$/**III**$m$), and all results are appended to Table 6 and related with the alkoxy-chain lengths $n$ of the acid component in Figure 8. As seen from Figure 8, independent of the length of the alkoxy-chains, the entropy values of N–I transitions ($\Delta S_{\text{N–I}}$) for **I**$n$/**II**$m$ and **I**$n$/**III**$m$ complexes are lower than $\Delta S_{\text{SmC-N}}/R$ transitions. The decrease that appeared in $\Delta S_{\text{N–I}}$ is reflected in the increment in the mesogenic core biaxiality, resulted in the flexible terminal alkoxy-chain, led to a decrease in the conformational entropy [51]. Non-correlation between the entropies and the terminal alkoxy-chain length may be due to the irregular change of lateral adhesion upon the increase of the total molecular length [51].

**Table 6.** Transition entropies ($\Delta S/R$), for the supramolecular complexes **I**$n$/**II**$m$, **I**$n$/**III**$m$, **IV**$n$/**II**$m$ and **IV**$n$/**III**$m$.

| System | $\Delta S_{\text{SmC-N}}/R$ | $\Delta S_{\text{N-I}}/R$ | System | $\Delta S_{\text{SmC-N}}/R$ |
|---|---|---|---|---|
| **I6/II8** | 0.9 | 0.3 | **IV6/II8** | 1.1 |
| **I8/II8** | 1.9 | 0.5 | **IV8/II8** | 1.3 |
| **I10/II8** | 1.6 | 0.5 | **IV10/II8** | 1.1 |
| **I12/II8** | 2.0 | 0.8 | **IV12/II8** | 1.1 |
| **I6/II16** | 1.4 | 0.4 | **IV6/II16** | 1.2 |
| **I8/II16** | 1.1 | 0.3 | **IV8/II16** | 1.4 |
| **I10/II16** | 1.2 | 0.4 | **IV10/II16** | 2.3 |
| **I12/II16** | 1.1 | 0.3 | **IV12/II16** | 2.4 |
| **I6/III8** | - | 0.6 | **IV6/III8** | - |
| **I8/III8** | 0.9 | 0.3 | **IV8/III8** | 0.3 |
| **I10/III8** | 1.3 | 0.7 | **IV10/III8** | 0.5 |
| **I12/III8** | 1.9 | 0.9 | **IV12/III8** | 0.4 |
| **I6/III16** | - | 0.5 | **IV6/III16** | - |
| **I8/III16** | 1.4 | 0.3 | **IV8/III16** | 0.5 |
| **I10/III16** | 1.5 | 0.7 | **IV10/III16** | 0.4 |
| **I12/III16** | 1.5 | 0.8 | **IV12/III16** | 2.2 |

Abbreviations: $\Delta S_{\text{SmC-N}}/R$ = smectic C to Nematic transition; $\Delta S_{\text{N-I}}/R$ = Nematic to isotropic liquid transition.

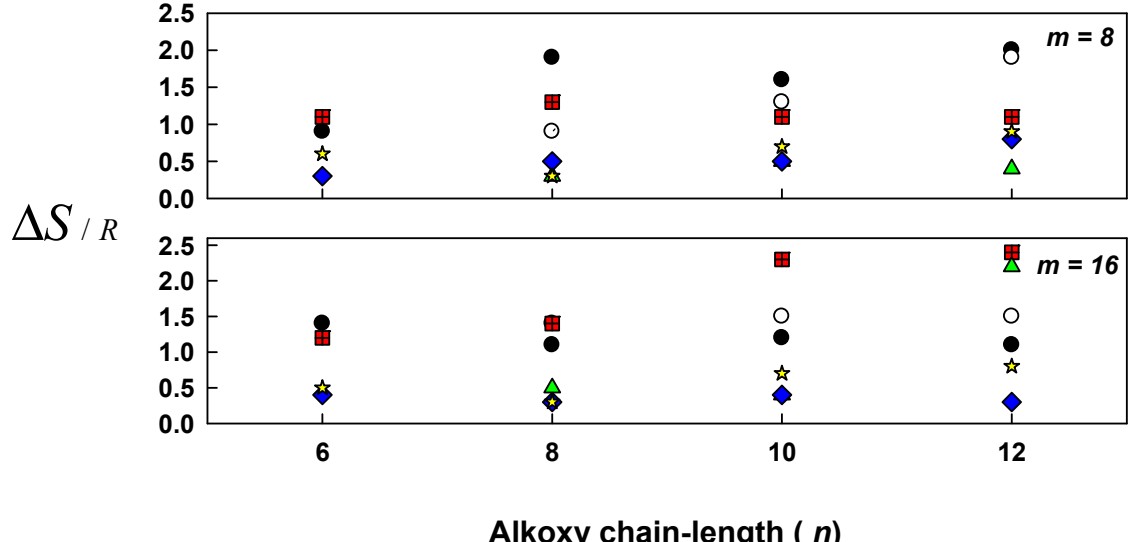

**Alkoxy chain-length ( *n*)**

**Figure 8.** Dependence of entropy change (Δ*S*/*R*) on the acid alkoxy chain length (*n*) of 1:1 supramolecular complexes: Smectic C-to-Nematic transition; **I***n*/**II***m* (●); **I***n*/**III***m* (○); **IV***n*/**II***m* (■); **IV***n*/**III***m* (▲) and Nematic-to-isotropic liquid transition; **I***n*/**II***m* (■); **I***n*/**III***m* (☆).

## 4. Conclusions

New series supramolecular hydrogen-bonded complexes with lateral groups were prepared and investigated by DSC and PLM. The investigated 1:1 complex was made by introducing a lateral $CH_3$ group by different spatial orientation angles into pyridine-based components. Individual compounds and their supramolecular complexes were confirmed by FTIR spectroscopy and computational calculations. The results revealed that all prepared complexes are enantiotropic dimorphic, exhibiting SmC and enhanced N phases. In addition, it was found that the incorporation of the phenylazo group increasing the length of the mesogenic moiety led to a broad nematic phase with high stability. DFT calculations for supramolecular complexes prepared were discussed. The results show that all compounds are polarizable while the complex **I12/III16** has more polarity than **I12/II16**. Moreover, the supramolecular H-boned complexes are softer than either the acid or the bases, so more polarizable than the free molecules.

**Supplementary Materials:** The following are available online at http://www.mdpi.com/2073-4352/9/3/133/s1, Table S1: Table S1: Optimized structures Cartesian coordinates of I*12*; Table S2: Optimized structures Cartesian coordinates of II*16*; Table S3: Optimized structures Cartesian coordinates of III*16*; Table S4: Optimized structures Cartesian coordinates of I*12*/II*16*; Table S5: Optimized structures Cartesian coordinates of I*12*/III*16*.

**Author Contributions:** H.A.A. and M.H. designed of the experiment and carried out the laboratory work; M.H., H.A.A. and O.A.A. accomplished the data analysis and drafted the manuscript; All authors gave final approval for publication.

**Funding:** This article is supported by the Deanship of Scientific Research, Taibah University (Research group No. 60333).

**Conflicts of Interest:** The authors declare no conflict of interest.

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
