# Peer review of "Phase Behavior and DFT Calculations of Laterally Methyl Supramolecular Hydrogen-Bonding Complexes"

_crystals, doi:10.3390/cryst9030133_

Round 1

Reviewer 1 Report

This contribution details both experimental and theoretical details on supramolecular complexes with methyl substitutions. As my area of expertise is in theoretical calculations I will focus my review on those aspects. The supramolecular systems investigated are relatively interesting (although the authors should modify the manuscript to highlight more of their significance, especially in the conclusions) and the computational methods used appear to be appropriate. However, the manuscript in its current form needs some significant work before it is suitable for publication in Crystals, or any other journal. I've separated my suggestions/queries into scientific and presentation categories below.

Scientific

1) The title is far too general for the study that has been carried out. This needs to be adjusted to reflect the actual systems investigated.

2) The references to the use of computational methods for designing molecules (lines 34&35, references 11-18) are far too focused on the previous work of the authors. This is a very active field and the references need to reflect this.

3) The B3LYP density functional should have its original citation in the methods section.

4) Which integration grid was used during the DFT calculations? This should be stated as a minimum of UltraFine grids should be used for calculations involving intermolecular interactions.

5) Optimized structures (in Cartesian coordinates) must be provided as supporting information.

6) Line 182 and Table 2. The temperature used in the calculation of the thermochemistry must be stated.

7) Lines 183-185 (and Table 2), sentence beginning "DFT calculations showed the extra stability of...". This is confusing as in Table 2 the delta H and delta G values for the species I12/III16 are higher than those of I12/II16 - suggesting they are less stable? This needs to be carefully checked. Additionally, the units of Hartree/particle seems too small here - why not use kJ/mol?

8) Line 186. The authors write "Moreover, all compounds showed polarizability". Where are the data for this? Polarizability and dipole moments are not the same thing - so either the text needs changing or the polarizabilities should be tabulated.

9) Table 2. Values for ZPVE are listed. Are these really Zero Point Vibrational Energy corrections? They are surely far too large for that. The authors need to be clear what these values are.

10) Line 199. A value of 1.38408 angstrom is given for the C=O distance in the free acid. This is given as 1.23711 in Table 3. I believe the value in the text is incorrect, but this must be checked carefully.

11) Lines 201-203: "Similarly, the azo group either for the acid moiety or the base moiety have been strengthened by decreasing the single bond character of the bond". I can't interpret what the authors mean with this sentence. What specifically of the azo group has been strengthened? How are the authors judging the amount of single bond character present? Do they simply mean that the N=N distance decreases on complex formation? If so, just say that.

12) Please define units for Table 3.

13) Are the DFT values in Table 4 scaled? The agreement with experiment appears to be remarkable for comparing harmonic approximation with experiment. The raw experimental spectra should also be provided as supporting information.

14) Line 227 and Fig 6. Please state the isosurface value used in plotting the MOs.

15) A key to the colors used should be included in the caption to Fig 7 (in addition to being in the main text). A color scale bar as part of the figure would be even better.

16) A significant part of the manuscript is devoted to computational aspects, but they make up only one simple line in the conclusions. This should be expanded upon to give readers some appreciation of the significance of the results.

Presentation

i) There are numerous places where characters in the text overlap, making reading difficult. This needs to be fixed.

ii) Line 51, towards the end of the line is "In", I believe this should be in bold as it refers to one of the chemical species.

iii) In numerous places the distance units are given as Ao. This needs to be replaced with the correct angstrom symbol, as A0 is used for distances in bohr.

iv) The standard of written English is not suitable for publication. This needs extensive editing beyond what can be expected from a peer-reviewer, but some of the more important points are highlighted below:

> Line 39, LCs needs to be defined.

> Line 215, "constituent" should probably be "consistent".

> Line 243 "H-boned" should be "H-bonded"

v) Lines 221 and 222: "Moreover, vibration peaks at 3150, 2545 , 1925 cm-1 due to Fermi resonance of A-type, B-type, C-type respectively." This isn't a full sentence and I don't understand what the authors mean.

Author Response

1) The title is far too general for the study that has been carried out. This needs to be adjusted to reflect the actual systems investigated.

The title has been addressed

2) The references to the use of computational methods for designing molecules (lines 34&35, references 11-18) are far too focused on the previous work of the authors. This is a very active field and the references need to reflect this.

Some references have been added

3) The B3LYP density functional should have its original citation in the methods section.

The citation has been included

4) Which integration grid was used during the DFT calculations? This should be stated as a minimum of UltraFine grids should be used for calculations involving intermolecular interactions.

The integration grid used during the DFT calculations has been stated in the manuscript

5) Optimized structures (in Cartesian coordinates) must be provided as supporting information.

The Cartesian coordinates have been included as supporting information.

6) Line 182 and Table 2. The temperature used in the calculation of the thermochemistry must be stated.

The temperature used in the calculation have been stated in the manuscript.

7) Lines 183-185 (and Table 2), sentence beginning "DFT calculations showed the extra stability of...". This is confusing as in Table 2 the delta H and delta G values for the species I12/III16 are higher than those of I12/II16 - suggesting they are less stable? This needs to be carefully checked. Additionally, the units of Hartree/particle seems too small here - why not use kJ/mol?

The data have been checked and the unites have been addressed

8) Line 186. The authors write "Moreover, all compounds showed polarizability". Where are the data for this? Polarizability and dipole moments are not the same thing - so either the text needs changing or the polarizabilities should be tabulated.

The text has been addressed

9) Table 2. Values for ZPVE are listed. Are these really Zero Point Vibrational Energy corrections? They are surely far too large for that. The authors need to be clear what these values are.

 Yes, ZPVE is Zero Point Vibrational Energy

10) Line 199. A value of 1.38408 angstrom is given for the C=O distance in the free acid. This is given as 1.23711 in Table 3. I believe the value in the text is incorrect, but this must be checked carefully.

The values have been checked and addressed

11) Lines 201-203: "Similarly, the azo group either for the acid moiety or the base moiety have been strengthened by decreasing the single bond character of the bond". I can't interpret what the authors mean with this sentence. What specifically of the azo group has been strengthened? How are the authors judging the amount of single bond character present? Do they simply mean that the N=N distance decreases on complex formation? If so, just say that.

The statement has been addressed

12) Please define units for Table 3.

The unites have been addressed

13) Are the DFT values in Table 4 scaled? The agreement with experiment appears to be remarkable for comparing harmonic approximation with experiment. The raw experimental spectra should also be provided as supporting information.

14) Line 227 and Fig 6. Please state the isosurface value used in plotting the MOs.

The isosurface value used in plotting the MOs has been stated

15) A key to the colors used should be included in the caption to Fig 7 (in addition to being in the main text). A color scale bar as part of the figure would be even better.

16) A significant part of the manuscript is devoted to computational aspects, but they make up only one simple line in the conclusions. This should be expanded upon to give readers some appreciation of the significance of the results.

The conclusion has been addressed

Presentation

i) There are numerous places where characters in the text overlap, making reading difficult. This needs to be fixed.

They have been addressed

ii) Line 51, towards the end of the line is "In", I believe this should be in bold as it refers to one of the chemical species.

It has been addressed

iii) In numerous places the distance units are given as Ao. This needs to be replaced with the correct angstrom symbol, as A0 is used for distances in bohr.

It has been addressed

iv) The standard of written English is not suitable for publication. This needs extensive editing beyond what can be expected from a peer-reviewer, but some of the more important points are highlighted below:

> Line 39, LCs needs to be defined.

It has been defined

> Line 215, "constituent" should probably be "consistent".

It has been addressed

> Line 243 "H-boned" should be "H-bonded"

It has been addressed

v) Lines 221 and 222: "Moreover, vibration peaks at 3150, 2545 , 1925 cm-1 due to Fermi resonance of A-type, B-type, C-type respectively." This isn't a full sentence and I don't understand what the authors mean.

It has been addressed

Reviewer 2 Report

Review attached.

Author Response

The manuscript is focused on co-crystalization of chosen acids and their details studied by a variety of methods. I have some serious comments for the text. Below the most important that influenced my understanding of the text as a reader.

Page 3, line 89. “Purity of all prepared compounds were checked with thinlayer

chromatography…’ While the compounds were purchased from Merck I do not get why to check their purity.

Only start materials were purchased from Merck but the final prepared compounds were synthesized according to the following references: Ahmed, H.; Naoum, M.; Saad, G., Mesophase behaviour of 1: 1 mixtures of 4-n-alkoxyphenylazo benzoic acids bearing terminal alkoxy groups of different chain lengths. Liquid Crystals 2016, 43, (9), 1259-1267; Naoum, M. M.; Fahmi, A. A.; Alaasar, M. A.; Salem, R. A., Supramolecular liquid crystals in binary and ternary systems. Thermochimica acta 2011, 517, (1-2), 63-73; Naoum, M. M.; Fahmi, A. A.; Mohammady, S. Z.; Abaza, A. H., Effect of lateral substitution on supramolecular liquid crystal associates induced by hydrogen-bonding interactions between 4-(4′-pyridylazo-3-methylphenyl)-4′′-alkoxy benzoates and 4-substituted benzoic acids. Liquid Crystals 2010, 37, (4), 475-486 and checked their purity by TLC.

On the other hand it could be that complexes were placed on TLC. In such a case the TLC is not suitable to look for 1:1 complexes. Also, I can not see the method of the synthesis of diaza compounds. It is worth noting that on page 2 line 66 the “4-n-alkoxy benzoic acids” were mentioned but not their “aza” derivatives. I suggest to be more specific.

The diaza compounds were synthesized according to the references mentioned in the previous point and we checked their purity by TLC, however the H-bonded complex formation was checked by FTIR and DSC results.

4-n-alkoxy benzoic acids was a typing mistake, it has been removed.

I also suggest to change the labeling. In some places the “In” looks like english “in” with capitalized first letter. This is especially true when “In” is not presented in bold font.

 All labels were checked and presented in bold font according to the Referees advice.

The methodology of calculations (for example the basis set) is repeated. This should be avoided.

The repeating has been avoided

The data in Table 2 do not tell me much. I’d rather see the energy of interaction between molecules or the energy of hydrogen bonding (based on QTAIM). The x,y,z constituents to the dipole moments do not tell much also especially when one can not see where the axis is.

The Cartesian coordinates have been included

The calculated bond lengths do not show the hydrogen bonding distances.

The calculated hydrogen bonding lengths have been included

Some literature is missing and thus, some parts of text should be rewritten to include the topic of the steric effect in supramolecular chemistry. Thus below some articles from one group only for consideration by the authors: J. Phys. Chem. A 2010, 114, 10421–10426, J. Phys. Chem. A 2010, 114, 12881–12887, Journal of Molecular Structure 1018 (2012) 84–87, J. Org. Chem. 2012, 77, 1653−1662, Journal of Molecular Structure 1054-1055 (2013) 157–163. Also, it si worth to search for more references that focus on steric effects in supramolecular chemistry but also in intramolecular hindering.

All references have been addressed according to the Referees suggestions.

At this stage I have to tell the manuscript, however interesting and worth reconsideration, is immature and I suggest to revise it.

Round 2

Reviewer 2 Report

Review attached as a PDF file.

Author Response

1.       I did not note it previously but ZPVE are much too large for those systems. The values given are, most probably, the sum of electronic and ZPVE.

Yes, it is sum of zero-point energy and electronic energy. But we have abbreviated it to be ZPVE.

Anyway, we have put the abbreviation at the footer of the Table 2

2. The sentence concluding the “non-linearity of the complexes” placed on page 12 is a bit trival. It is easy to predict that when one realizes the number of sp3 hybridized atoms.

Yes, we know that they ar non linear due to the SP3 hybridization but we want to show that the H-bonding increases the non linearity.

3. Page 13: it is “On the other hands” while it should be “hand”.

It has been addressed.

4. For the bond lengths the Angsroem unit should be used (circle above letter A) or the values should be given in pm.

It has been addressed.

5. Page 14: was the experimental IR recorded in gas phase? The sentence suggests that.

The experimental IR phase addressed to be in KBr

6. I do not understand why the oxygen atom in studied compounds or complexes shall have the least electron density since it is the most electronegative atom in those structures. Even more, the MEP and the description in text tells that oxygen or nitrogen are less prone to protonation than other atoms. This is somehow reversed.

It has been revised.